# Limited Genetic Diversity of blaCMY-2-Containing IncI1-pST12 Plasmids from Enterobacteriaceae of Human and Broiler Chicken Origin in The Netherlands

**DOI:** 10.3390/microorganisms8111755

**Published:** 2020-11-08

**Authors:** Evert P. M. den Drijver, Joep J. J. M. Stohr, Jaco J. Verweij, Carlo Verhulst, Francisca C. Velkers, Arjan Stegeman, Marjolein F. Q. Kluytmans-van den Bergh, Jan A. J. W. Kluytmans

**Affiliations:** 1Microvida, Laboratory for Medical Microbiology and Immunology, Elisabeth-TweeSteden Hospital, 5000 AS Tilburg, The Netherlands; joep.stohr@gmail.com (J.J.J.M.S.); j.verweij@etz.nl (J.J.V.); 2Microvida, Laboratory for Microbiology, Amphia Hospital, 4818 CK Breda, The Netherlands; cverhulst@amphia.nl (C.V.); jankluytmans@gmail.com (J.A.J.W.K.); 3Faculty of Veterinary Medicine, Department of Farm Animal Health, Utrecht University, 3584 CL Utrecht, The Netherlands; F.C.Velkers@uu.nl (F.C.V.); J.A.Stegeman@uu.nl (A.S.); 4Department of Infection Control, Amphia Hospital, 4818 CK Breda, The Netherlands; marjoleinkluytmans@gmail.com; 5Amphia Academy Infectious Disease Foundation, Amphia Hospital, 4818 CK Breda, The Netherlands; 6Julius Center for Health Sciences and Primary Care, University Medical Center Utrecht, Utrecht University, 3508 GA Utrecht, The Netherlands

**Keywords:** AmpC β-lactamase, plasmid, IncI1, blaCMY-2

## Abstract

Distinguishing epidemiologically related and unrelated plasmids is essential to confirm plasmid transmission. We compared IncI1–pST12 plasmids from both human and livestock origin and explored the degree of sequence similarity between plasmids from Enterobacteriaceae with different epidemiological links. Short-read sequence data of Enterobacteriaceae cultured from humans and broilers were screened for the presence of both a *bla*_CMY-2_ gene and an IncI1–pST12 replicon. Isolates were long-read sequenced on a MinION sequencer (OxfordNanopore Technologies). After plasmid reconstruction using hybrid assembly, pairwise single nucleotide polymorphisms (SNPs) were determined. The plasmids were annotated, and a pan-genome was constructed to compare genes variably present between the different plasmids. Nine *Escherichia coli* sequences of broiler origin, four *Escherichia coli* sequences, and one *Salmonella enterica* sequence of human origin were selected for the current analysis. A circular contig with the IncI1–pST12 replicon and *bla*_CMY-2_ gene was extracted from the assembly graph of all fourteen isolates. Analysis of the IncI1–pST12 plasmids revealed a low number of SNP differences (range of 0–9 SNPs). The range of SNP differences overlapped in isolates with different epidemiological links. One-hundred and twelve from a total of 113 genes of the pan-genome were present in all plasmid constructs. Next generation sequencing analysis of *bla*_CMY-2_-containing IncI1–pST12 plasmids isolated from Enterobacteriaceae with different epidemiological links show a high degree of sequence similarity in terms of SNP differences and the number of shared genes. Therefore, statements on the horizontal transfer of these plasmids based on genetic identity should be made with caution.

## 1. Introduction

Antimicrobial resistance in Gram-negative bacteria is a worldwide growing public health problem [1,2]. The gut is an important reservoir for resistant Gram-negative bacteria, both in humans and livestock [3,4]. Antimicrobial resistance in livestock has been suggested as a potential source for resistance in humans, with a growing number of studies published on this potential transmission route for antimicrobial resistance mechanisms in Gram-negative bacteria [5,6,7]. AmpC beta-lactamase-production is an example of these mechanisms as a potential source for 3rd generation cephalosporin resistance in Gram-negative bacteria [8].

Plasmids are an important vector for antimicrobial resistance dissemination with genes for various resistance mechanisms (e.g., AmpC beta-lactamase genes) being located on these mobile genetic elements. Incompatibility group I1 (IncI1) plasmids of the plasmid sequence type (pST) 12 have been associated with the spread of *bla*_CMY-2_, which is the most common AmpC beta-lactamase gene [9,10,11]. Recent studies show that the sequence of IncI1 plasmids is highly conserved [12,13,14,15,16]. Most studies to date are based on short-read sequence mechanisms [13,14,16]. However, it remains challenging to study plasmid transmission using short-read sequencing data alone. Repeated sequences, often shared between plasmid and chromosomal DNA, hinder the assembly of the bacterial genome from short-read data, often resulting in contigs of which the origin, either plasmid or chromosomal, cannot be resolved [17]. This limits the interpretation of plasmid transmission by not providing accurate prediction of the total plasmid sequence. Recently, a combination of short- and long-read sequence data provided an accurate analysis, such as shown in a recent study on IncI1 plasmids of pST3 and pST7 [15]. Everything considered, the amount of studies using combined short- and long-read sequencing data of IncI1-pST12 plasmids from human and livestock origin is still limited. The transmission of antimicrobial resistant bacteria within and between domains is predominantly based on the comparison of bacterial chromosome. However, when only typing the bacterial chromosome, the transmission of resistance gene-containing plasmids can go undetected. Although plasmid replicon typing combined with pMLST data can be useful to monitor the spread of plasmids through populations, more accurate distinguishing of related from non-related plasmids based on molecular characteristics (e.g., number of single nucleotide polymorphisms (SNP) differences) is essential for using sequence data to detect plasmid transmission. We hypothesize that a combination of short- and long-read sequence data of *bla*_CMY-2_ containing IncI1–pST12 plasmids reveal highly conserved plasmid sequencing, which complicates distinguishing plasmid transmission between epidemiologically related and unrelated isolates. The objective of the current study is to determine the relatedness between IncI1–pST12 plasmids of epidemiologically related and unrelated Enterobacteriaceae isolates from humans and livestock, and we explore the possibility of accurately distinguishing related from unrelated samples based on plasmid sequencing data alone.

## 2. Materials and Methods

### 2.1. Collection of Isolates

#### 2.1.1. AmpC *E. coli* Isolates from i-4-1-Health Dutch-Belgian Cross-Border Project

As part of the i-4-1-Health project, human and broiler samples were collected as described by Kluytmans-van den Bergh et al. [18]. After vortexing, the nylon-flocked swabs in 2 mL Cary–Blair medium (FecalSwab^®^, Copan Italy, Brescia, Italy) were plated on a blood agar plate (growth control, performed since 2011), and the liquid Cary–Blair medium was mixed in tryptic soy broth (TSB) and incubated for 18–24 h (35–37 °C). Broths were subcultured on an AmpC selective MacConkey agar containing cefotaxime and cefoxtin (1 and 8 mg/mL, respectively) on half the plate and ceftazidime and cefoxitin (1 and 8 mg/mL, respectively) on the other half of the plate (Mediaproducts, Groningen, Germany) [19]. For all oxidase-negative isolates that grew on either side of the selective agar plates, species identification was performed by automated mass spectrometry systems (VitekMS, bioMérieux, Marcy l’Etoile, France). Susceptibility testing was performed using Vitek 2 (bioMérieux, Marcy l’Etoile, France). The presence of AmpC in all oxidase-negative isolates was phenotypically confirmed using the D68C AmpC & ESBL Detection Set (Mastdiscs, Mastgroup Ltd., Bootle, UK) and interpreted according to the manufacturer’s instructions. All phenotypically confirmed isolates were sequenced using an Illumina MiSeq sequencer (Illumina, San Diego, CA, USA). DNA isolation and sequencing were performed as described by Coolen et al. [20]. De novo assembly and error correction were performed using SPAdes version 3.9.1 [21].

#### 2.1.2. AmpC *E. coli* Isolates from Amphia Prevalence Screening

*pampC* gene containing *E. coli* isolates were selected from a prevalence screening, which had been performed in the Amphia hospital described by Den Drijver et al. [22]. Rectal swabs taken from hospital patients were pre-enriched using selective TSB containing cefotaxime (0.25 mg/L) and vancomycin (8 mg/L) and subsequently cultured on a MacConkey agar plate containing cefotaxime (1 mg/L) or a MacConkey double agar plate containing cefotaxime and cefoxtin (1 and 8 mg/mL, respectively) on half the plate and ceftazidime and cefoxitin (1 and 8 mg/mL, respectively) on the other half of the plate (Mediaproducts, Groningen) [19]. For all oxidase-negative isolates that grew on either side of the selective agar plates, species identification was performed by automated mass spectrometry systems (VitekMS, bioMérieux, Marcy l’Etoile, France). Susceptibility testing was performed using Vitek 2 (bioMérieux, Marcy l’Etoile, France). The presence of AmpC in all oxidase-negative isolates was phenotypically confirmed using the D68C AmpC & ESBL Detection Set (Mastdiscs, Mastgroup Ltd., Bootle, UK) and interpreted according to the manufacturer’s instructions. All phenotypically confirmed isolates were sequenced in the University of Groningen Medical Center (UMCG) using MiSeq (Illumina, San Diego, CA, USA) and assembled with CLC Genomics Workbench 9.0, 9.0.1 or 9.5.2 (Qiagen, Hilden, Germany) as was previously described in more detail by Kluytmans-van den Bergh et al. [23].

#### 2.1.3. pAmpC-encoding Clinical Isolates from Elisabeth-Tweesteden Hospital

Suspected *pampC* gene containing *E. coli* isolates from blood cultures were selected retrospectively from our laboratory database based upon the presence of a phenotype (cefoxitin minimal inhibitory concentration (MIC) > 8 mg/L and/or cefotaxime MIC ≥ 1mg/L and/or ceftazidime MIC ≥ 1mg/L. One *Salmonella enterica* serotype Kentucky isolate from a fecal sample was selected from our laboratory database based upon the presence of an AmpC suspected phenotype (cefoxitin MIC > 8 mg/L and/or cefotaxime MIC ≥ 1mg/L and/or ceftazidime MIC ≥ 1mg/L). The isolates were recultured from deep frozen samples on blood agar and identified using the MALDI-TOF MS (BD Diagnostic Systems, Sparks, MD, USA). Susceptibility testing was performed using a Phoenix Automated Microbiology System (BD Diagnostic Systems, Sparks, MD, USA). The isolates were sequenced using an Illumina MiSeq sequencer (Illumina, San Diego, CA, USA). DNA isolation and sequencing were performed as described by Coolen et al. [20]. De novo assembly and error correction were performed using SPAdes version 3.9.1 [21].

### 2.2. Whole-Genome Bioinformatics Analysis of Short-Read Sequencing Data

The presence of acquired resistance genes was identified by uploading assembled genomes to the ResFinder web service of the Center for Genomic Epidemiology (version 3.1) [24]. The presence of plasmid replicons and the typing of a specific IncI plasmid was performed using pMLST (version 2.0) [25]. The genomes were selected based on a 100% match to *bla*_CMY-2_ and IncI-pST12. Typing of a specific multi locus sequence type (MLST) was performed using the MLST web service of the Center for Genomic Epidemiology (version 2.0), and *fim* typing was performed using FimTyper (version 1.0), Center for Genomic Epidemiology [26,27].

### 2.3. Long-Read Sequencing and Hybrid Assembly

No more than two isolates of the same flock or patient belonging to the same MLST were selected for further long-read sequencing.

All isolates were long-read sequenced on a MinION sequencer using the FLO-MIN106D flow cell and the Rapid Barcoding Sequencing Kit SQK RBK004 according to the standard protocol provided by the manufacturer (Oxford Nanopore Technologies, Oxford, UK). A hybrid assembly of long-read and short-read sequence data was performed using Unicycler v.0.8.4 [28]. Whole-genome MLST (wgMLST) (core and accessory genome) was performed for all isolates using Ridom SeqSphere+, version 4.1.9 (Ridom, Münster, Germany). Species-specific wgMLST typing schemes were used as described previously [23]. The pairwise genetic difference between isolates of the same species was calculated by dividing the total number of allele differences by the total number of shared alleles from the typing scheme present in both sequences, using a pairwise ignoring missing values approach. Genetic relatedness was determined using the thresholds for wgMLST-based genetic distance of 0.0095, as described previously [23].

### 2.4. Plasmid Analysis

The genomes created using the hybrid assembly were uploaded to the online bioinformatics tools ResFinder v.2.1, VirulenceFinder v.1.2 and PlasmidFinder v.1.2. (Center for Genomic Epidemiology, Technical University of Denmark, Lingby, Denmark) [24,25,29]. Circular components created by the hybrid assembly that were smaller than 1000 kb and that contained an IncI1-pST12 plasmid replicon and a *bla*_CMY-2_ gene were extracted from the assembly graph using BANDAGE v0.8.1. [30]. All extracted plasmid components were annotated using Prokka v1.13.3 [31]. Using snippy v4.4.59 (https://github.com/tseemann/snippy), the number of single nucleotide polymorphisms (SNPs) was determined between the extracted plasmid components using a *bla*_CMY-2_ gene containing IncI1–pST12 plasmid extracted from the GenBank (accession number: MH472638.1) as reference [12]. A pan-genome was constructed, and a gene presence or absence was determined for all extracted plasmid components using roary v3.12 [32]. All extracted plasmids consisting of a single circular contig were aligned using GView 1.7 [33] and progressiveMAUVE v2.4.0 to detect possible rearrangements [34]. If a hypervariable region is identified, the sequence of this region and its flanking regions are extracted using biopython v1.37. Moreover, segments (A, B, C, D) and flanking genes (PilV and rci) of a previously described hypervariable shufflon region of the IncI1 replicon containing plasmids (GenBank accession nr: AB027308.1) were BLAST searched in the extracted hypervariable regions [35,36].

### 2.5. Classification of Pairwise Comparisons

Pairwise comparisons of assembled plasmids were classified according to the known epidemiological link between the isolates: (i) same sample; (ii) same ward/flock but different sample; (iii) same location (hospital or farm) but different ward/flock and sample; (iv) same domain (human or broiler) but different location, ward/flock and sample; and (v) no known epidemiological link, i.e., different domain, location, ward/floc, and sample.

### 2.6. Ethical Statement

The I-4-1-Healt study was judged to be beyond the scope of the Dutch Medical Research Involving Human Subjects Act and the Belgian Law on Experiments on Humans, dated 7 May 2004. Written or verbal informed consent for data collection and taking a fecal, perianal, or gastrointestinal stoma swab for microbiological culture is obtained from all participants or their legal representatives. For the veterinary domain, approval by an animal welfare body is not required. All human data are anonymized, i.e., data cannot be directly or indirectly related to their source. Data on institutions and farms are pseudonymized, i.e., identifying information is replaced by a code, and a key file that links this code to the identifying information is kept separate from the research data.

## 3. Results

### 3.1. Isolate Characteristics

A total of 2508 human cases from four different hospitals and 119 broilers from 14 different farms were screened for the presence of plasmid encoded AmpC genes, e.g *bla*_CMY-2_ (Appendix A). In 107 isolates, an AmpC phenotype was confirmed based on the D68C AmpC & ESBL Detection Set. Sixteen of 107 isolates contained both an IncI1 pST12 and a *bla*_CMY-2_ gene (Appendix A). Based upon the above-mentioned selection criteria, fourteen isolates were included for long-read sequencing analysis, i.e., thirteen *E. coli* and one *Salmonella enterica*, serotype Kentucky (Table 1). Nine of the *E. coli* isolates were from one broiler farm; the other isolates were from human origin. The *E. coli* isolates included five different MLSTs and *fim* types. Based on wgMLST analysis, four different clusters could be identified (Figure 1, Table 1, and Appendix A). Additional information regarding antimicrobial resistance phenotype and genotype of the included isolates is provided in Appendix A.

### 3.2. Plasmid Analysis

In the hybrid assembly of fourteen sequences, both the IncI1–pST12 replicon gene and *bla*_CMY-2_ gene were located on a single circular contig ranging in size from 98,410 to 98,999 bp. No additional antimicrobial resistance or virulence genes were detected on any of the extracted plasmids. The number of SNP’s detected between the fourteen plasmids ranged from zero to nine SNPs (Table 2). When comparing the plasmids extracted from the selected isolates to a publicly available IncI1–pST12 *bla*_CMY-2_ gene-containing plasmid extracted from the GenBank (accession number: MH472638.1), the number of SNPs detected ranged from 0 to 7 (Table 2). A small SNP difference was seen between epidemiologically related strains with a maximum difference of two SNPs. The range of SNP differences overlapped between epidemiologically related and unrelated plasmids (Table 3). The median number of SNP differences of plasmids in a different domain or different location, but the same domain was higher than in the other three pairwise comparison groups.

The total number of genes detected in the fourteen plasmids was 113, of which 112 were detected in all plasmids. One gene was present only in one plasmid (pEC11) and encoded for a hypothetical protein. An alignment of coding regions of the fourteen plasmids revealed no rearrangements between the described plasmids (Figure 2). However, progressive MAUVE alignment of non-coding regions revealed a small highly variable region of 519 to 1096 bp in all plasmids (Appendix A). This hypervariable region and approximately 2125 bp of the flanking sequence were extracted from all plasmids. The genes *PilV* and *rci* were detected in the flanking regions of the hypervariable region of all plasmids (Table 4). Moreover, in all plasmids, either one (B) or two (A, B) shufflon segments were detected in the extracted hypervariable region of the various plasmids (Table 4). No rearrangements were detected in any of the other regions.

## 4. Discussion

The current study included *E. coli* isolates of various sequence types and a *S. enterica* isolate, which were from both human and broiler origin. Plasmid analysis based on short- and long-read sequence data of *bla*_CMY-2_ containing IncI1-pST12 plasmids from the included isolates revealed a low number of SNP differences and a high number of shared genes between the various plasmids extracted. Despite the tendency of median SNP increase from epidemiologically related to unrelated plasmids, the range in number of SNPs detected overlapped between every classified epidemiological link in the current study. A small SNP difference was seen between epidemiologically related strains with a maximum difference of two SNPs. Furthermore, only one gene was variably present between the different plasmids, and no rearrangements were observed apart from a small, highly variable region. This area is the formerly described highly variable shufflon region at the C-terminal end of the PilV protein [35,36].

A high degree of similarity between IncI1–pST12 plasmids was previously reported [12,13,14,16]. However, all of the studies either contained only plasmids extracted from one *E. coli* sequence type (ST131) [12] or the included plasmids were primarily of poultry origin [14]. All of the studies used either gene presence/absence-based or SNP-based analysis, but not both, possibly missing subtle differences between various plasmids. Shirakawa et al. used a combination of short-read sequence data of different *bla*_CMY-2_-containing plasmids from Japanese poultry and human origin, together with plasmid sequence data retrieved from the National Center for Biotechnology Information nucleotide database (https://www.ncbi.nlm.nih.gov/) to perform an extensive plasmid comparative analysis. Their clustering analysis showed a high similarity among the IncI1–pST12 plasmids as well; however, this study did not provide further detail on the SNP differences of possible rearrangements within the plasmid sequences. Moreover, these studies predominately used in silico reference-based plasmid reconstructions of short-read sequence data rather than performing a hybrid assembly of both short- and long-read sequence data. A recent study by Valcek et al. on IncI1–pST3 and IncI1–pST7 plasmids showed that using combined long-read and short-read sequencing data improves the accuracy of a full plasmid analysis, e.g., of rearrangements [15]. The current study is the first study describing plasmid differences using both gene presence/absence-based and SNP-based analysis. Moreover, rearrangements between the different plasmids could be detected such as those shown in the hypervariable region, which were missed in previous studies based on only short-read sequences.

Several studies have described outbreaks with *bla*_CMY-2_-harboring Enterobacteriaceae [37,38,39,40]. Since the *bla*_CMY-2_ is predominantly located on plasmids, horizontal transfer of the plasmid in an outbreak can go undetected if only typing of the bacterial chromosome is performed. Distinguishing epidemiologically related and unrelated plasmids is essential to confirm plasmid transmission in an outbreak. Therefore, statements on the horizontal transfer of these plasmids based on genetic identity should be made with caution. However, given the conservation of the IncI1–pST12 plasmids, they could instead be used as a tool to monitor the speed and breadth of spread of these plasmids through populations, either different in place of origin or bacterial host.

The current study is the first to explore *bla*_CMY-2_-containing IncI1–pST12 plasmids from related and unrelated isolates, using combined short- and long-read sequencing data. Moreover, this study includes isolates from different species, sequence types, and domains, both from human and broiler origin. Two different comparison techniques, either gene presence/absence and SNP differences, were used. Furthermore, combining long-read and short-read sequence data provided full plasmid analysis, including the presence of rearrangements.

By combining the isolate collections from three different studies, we screened a relatively large amount of human and broiler cases. However, due to low prevalence of *bla_CMY-2_* in the Netherlands, our sample size remained relatively small. This results in the main limitation of the current study, as the small sample size precludes the use of statistical test and caution must be applied, as the findings should be confirmed in a study with a larger sample size. Preferably, such a study should include isolates of different species, sequence types, and origin of isolation containing IncI1–pST12 plasmids. Furthermore, the current study only included plasmids of broilers isolated in one farm; therefore, other plasmids of veterinary origin should be added in future studies to confirm our findings.

In conclusion, IncI1–pST12 plasmids of epidemiologically related and unrelated Enterobacteriaceae of both human and broiler origin in the current explorative study show a high degree of sequence similarity in terms of SNP differences and the number of shared genes.

## Figures and Tables

**Figure 1 microorganisms-08-01755-f001:**
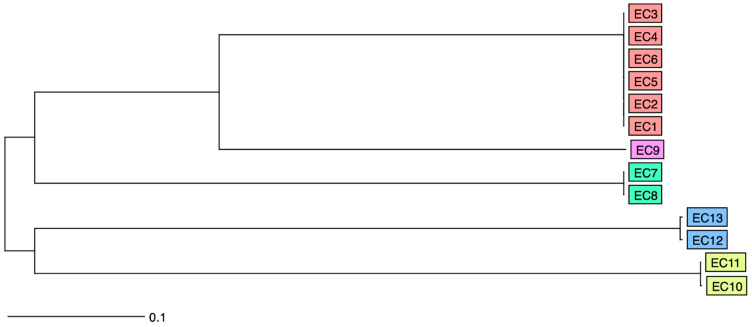
Neighbor-joining tree representing the whole-genome multi locus sequence type (wgMLST) analysis of the different *E. coli* isolates included in the study. Isolates belonging to the same clonal clusters are represented in the identical colors.

**Figure 2 microorganisms-08-01755-f002:**
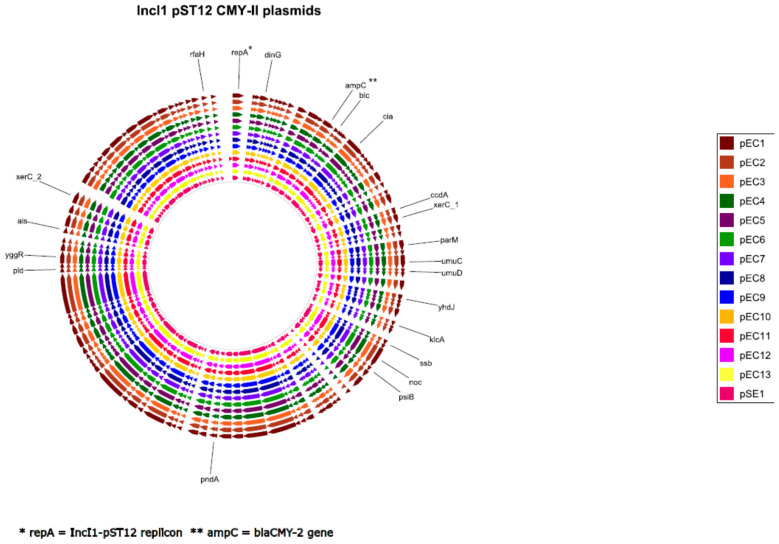
GView alignment of the various plasmid sequences. Each arrow represents a coding sequence and not necessarily transcriptional direction; gene names are depicted as generated by prokka.

**Table 1 microorganisms-08-01755-t001:** Descriptive characteristics of fourteen IncI1–pST12 and *bla*_CMY-2_ containing isolates.

Isolate	Species	Multilocus ST ^a^	wgMLST Cluster	Fim	Origin	Sample Location	Flock or Ward	Sample Source	Month and Year of Isolation	Accession No.
EC1	*E. coli*	ST665	1	fimH30	Broiler	Farm 1	Flock 1	Fecal swab 1	Nov 2017	ERS4591617
EC2	*E. coli*	ST665	1	fimH30	Broiler	Farm 1	Flock 1	Fecal swab 2	Nov 2017	ERS4591618
EC3	*E. coli*	ST665	1	fimH30	Broiler	Farm 1	Flock 2	Fecal swab 3	Nov 2017	ERS4591619
EC4	*E. coli*	ST665	1	fimH30	Broiler	Farm 1	Flock 2	Fecal swab 4	Nov 2017	ERS4591620
EC5	*E. coli*	ST665	1	fimH30	Broiler	Farm 1	Flock 3	Fecal swab 5	Nov 2017	ERS4591621
EC6	*E. coli*	ST665	1	fimH30	Broiler	Farm 1	Flock 3	Fecal swab 6	Nov 2017	ERS4591622
EC7	*E. coli*	ST86	2	fimH289	Broiler	Farm 1	Flock 3	Fecal swab 5	Nov 2017	ERS4591623
EC8	*E. coli*	ST86	2	fimH289	Broiler	Farm 1	Flock 3	Fecal swab 7	Nov 2017	ERS4591624
EC9	*E. coli*	ST6856		fimH71	Broiler	Farm 1	Flock 3	Fecal swab 6	Nov 2017	ERS4591625
EC10	*E. coli*	ST131	3	fimH22	Human	Hospital 1	Ward 1	Blood 1	Oct 2013	ERS4591626
EC11	*E. coli*	ST131	3	fimH22	Human	Hospital 2	Ward 1	Blood 2	Jul 2014	ERS4591627
EC12	*E. coli*	ST973	4	fimH95	Human	Hospital 3	Ward 1	Rectal swab 1	Dec 2017	ERS4591628
EC13	*E. coli*	ST973	4	fimH95	Human	Hospital 3	Ward 2	Rectal swab 2	Dec 2017	ERS4591629
SE1	*Salmonella enteritidis*	-		-	Human	Primary care unit	n.a.	Feces	Aug 2018	ERS4591630

^a^ Multilocus Sequence Type (ST) according to Enterobase (http://enterobase.warwick.ac.uk/).

**Table 2 microorganisms-08-01755-t002:** Number of single nucleotide polymorphisms (SNPs) detected between the 14 extracted plasmids and GenBank reference plasmid MH472638.1.

	pEC1	pEC2	pEC3	pEC4	pEC5	pEC6	pEC7	pEC8	pEC9	pEC10	pEC11	pEC12	pEC13	pSE1	MH472638.1
pEC1	0	2	2	2	2	2	3	3	3	3	3	9	8	6	2
pEC2	2	0	0	0	0	0	1	1	1	1	1	7	6	4	0
pEC3	2	0	0	0	0	0	1	1	1	1	1	7	6	4	0
pEC4	2	0	0	0	0	0	1	1	1	1	1	7	6	4	0
pEC5	2	0	0	0	0	0	1	1	1	1	1	7	6	4	0
pEC6	2	0	0	0	0	0	1	1	1	1	1	7	6	4	0
pEC7	3	1	1	1	1	1	0	0	0	2	2	8	7	5	1
pEC8	3	1	1	1	1	1	0	0	0	2	2	8	7	5	1
pEC9	3	1	1	1	1	1	0	0	0	2	2	8	7	5	1
pEC10	3	1	1	1	1	1	2	2	2	0	0	8	7	5	1
pEC11	3	1	1	1	1	1	2	2	2	0	0	8	7	5	1
pEC12	9	7	7	7	7	7	8	8	8	8	8	0	1	5	7
pEC13	8	6	6	6	6	6	7	7	7	7	7	1	0	4	6
pSE1	6	4	4	4	4	4	5	5	5	5	5	5	4	0	4
MH472638.1	2	0	0	0	0	0	1	1	1	1	1	7	6	4	0

**Table 3 microorganisms-08-01755-t003:** Median and range of SNP differences in pairwise comparisons per epidemiological link.

		SNP Differences	
	*n* of Pairwise Comparisons	Median	Range
Same sample	2	1	1
Same flock, different sample	10	0.5	0–2
Same location, different ward/flock	25	1	0–3
Same domain, different location	9	5	0–8
Different domain	45	4	1–9

**Table 4 microorganisms-08-01755-t004:** Shufflon segments in variable regions of the different plasmids included (direction: ′5–′3).

Plasmid	Shufflon Segments
pEC1	*PilV*	A	B	*rci*
pEC2	*PilV*	A	B	*rci*
pEC3	*PilV*	A	B	*rci*
pEC4	*PilV*	B	*rci*	
pEC5	*PilV*	B	*rci*	
pEC6	*PilV*	A	B	*rci*
pEC7	*PilV*	B	*rci*	
pEC8	*PilV*	A	B	*rci*
pEC9	*PilV*	A	B	*rci*
pEC10	*PilV*	A	B	*rci*
pEC11	*PilV*	B	A	*rci*
pEC12	*PilV*	B	*rci*	
pEC13	*PilV*	B	*rci*	
pSE1	*PilV*	B	*rci*

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
