# Peer review of "Limited Genetic Diversity of blaCMY-2-Containing IncI1-pST12 Plasmids from Enterobacteriaceae of Human and Broiler Chicken Origin in The Netherlands"

_microorganisms, 2020, doi:10.3390/microorganisms8111755_

Round 1

Reviewer 1 Report

The major points raised during revision of first submission, have been satisfactorily addressed in this resubmission

Author Response

Dear Editors, dear Reviewer,

We thank the reviewer for his/her comments on the manuscript, taking into account all time and effort put in this review and we are happy to hear that the major points raised during revision of the first submission have been satisfactorily addressed in the resubmission

Reviewer 2 Report

Overview Summary:

The work by den Drijver et al. describes the sequencing of 107 isolates from human and broiler origin and then subsequent analysis of strains containing IncI1 pST12 and a blaCMY-2 gene.

Conceptual Comments:

  1. I still think that the authors could put in 1-2 well-placed sentences that outline directly how their results were comparable or in contrast to other studies that may have had less analyses (references 12-16 in particular). They do mention that the other studies may have missed key findings due to their abbreviated analyses, but a clearer description of what the present results found over past work should be evident.

  1. Disscussion – The authors appropriately mention that the high sequence similarity among IncI1-pST12 plasmids should be used with caution when determining plasmid transmission among relevant isolates. However, a sentence or two would be appropriate that suggest that given this conservation, the IncI1-pST12 plasmids could instead be used as a tool to monitor the speed and breadth of spread of these genes/plasmids through populations (either distant in phylogeny or geographical location). This would certainly add value to the discussion and interpretation of results.

  1. As mentioned in the manuscript, the current study would tremendously benefit from a larger sample size (as they recognize in the conclusions). Isolates from different origins would greatly enhance their analysis. At the very least, isolates from different farms should have been analyzed, but the interrogative and statistical power would be even better if different geographical locations, phenotypes, animal sources and/or species were included in their analysis. This is the greatest shortcoming of this manuscript but I understand that it may be beyond the scope of this manuscript to address this issue at this point.

Grammar/Editorial/Minor Comments:

Abstract – Bacterial names need italicizing.

Throughout document – check for extra spaces where text has been changed from the last version.

L56 – reword “sequence provided an accurate analysis, as was shown in a”

L63 – reword “accurate distinguishing of related”

L75 – reword “nylon-flocked swabs in 2 ml”

L77 – clarify – “liquid Cary-Blair medium was inoculated” was the media inoculated or the cells? Typically media would be mixed and cells would be inoculated.

L163 – The addition of numbers here is much better and frames the study nicely

L170 – reword “could be identified”

L184 – reword “Table 2). A small”

L192/ 194 – 14 written as a number

L197 – reword “hypervariable region and the flanking sequence”. Approximately how much in bp?

L197/198 – “genes PilV and rci” – if these are genes, then they need to be written in italics – pilV?

L222 – reword “contained plasmids from Japanese”

L225 – reword “did not provide”

L231 – reword “[15]. All of the studies used”

L245 – reword “collections from three different studies we screened relatively large amounts”

Table S1 – The title is still too long and cumbersome. I would suggest a shorter informative title and then make use of a legend below the table to supply extra details.

Author Response

Dear Editors, dear Reviewer,

We thank the reviewer for his/her comments on the manuscript, taking into account all time and effort put in this review. We have revised the comments of the reviewer and attached a revision of the manuscript with track changes.

Conceptual Comments:

  1. I still think that the authors could put in 1-2 well-placed sentences that outline directly how their results were comparable or in contrast to other studies that may have had less analyses (references 12-16 in particular). They do mention that the other studies may have missed key findings due to their abbreviated analyses, but a clearer description of what the present results found over past work should be evident.

We agree with the reviewer and have added the following sentences to the discussion section: 

"The current study is the first study describing plasmid differences using both gene presence/absence based and SNP based analysis. Moreover, rearrangements between the different plasmids could be detected such as shown in the hypervariable region, that were missed in previous studies based on only short-read sequences." (lines 242-245)

Hopefully this clarifies the added value of our study compared to prior analyses.

  1. Discussion – The authors appropriately mention that the high sequence similarity among IncI1-pST12 plasmids should be used with caution when determining plasmid transmission among relevant isolates. However, a sentence or two would be appropriate that suggest that given this conservation, the IncI1-pST12 plasmids could instead be used as a tool to monitor the speed and breadth of spread of these genes/plasmids through populations (either distant in phylogeny or geographical location). This would certainly add value to the discussion and interpretation of results.

We agree with the reviewer and thank him/her for the suggestion. We added the following sentence to the discussion section:

"However, given the conservation of the IncI1-pST12 plasmids, they could instead be used as a tool to monitor the speed and breadth of spread of these plasmids through populations, either different in place of origin or bacterial host." (lines 251-253)

  1. As mentioned in the manuscript, the current study would tremendously benefit from a larger sample size (as they recognize in the conclusions). Isolates from different origins would greatly enhance their analysis. At the very least, isolates from different farms should have been analyzed, but the interrogative and statistical power would be even better if different geographical locations, phenotypes, animal sources and/or species were included in their analysis. This is the greatest shortcoming of this manuscript but I understand that it may be beyond the scope of this manuscript to address this issue at this point.

We acknowledge the limitations mentioned here by the reviewer. As described in the discussion, we are hopeful that future studies may resolve this problem (lines 262-268). 

Grammar/Editorial/Minor Comments:

Abstract – Bacterial names need italicizing.

Revised according to the suggestion of the reviewer

Throughout document – check for extra spaces where text has been changed from the last version.

Extra spaces have been removed as suggested by the reviewer

L56 – reword “sequence provided an accurate analysis, as was shown in a”

Revised according to the suggestion of the reviewer

L63 – reword “accurate distinguishing of related”

Revised according to the suggestion of the reviewer

L75 – reword “nylon-flocked swabs in 2 ml”

Revised according to the suggestion of the reviewer

L77 – clarify – “liquid Cary-Blair medium was inoculated” was the media inoculated or the cells? Typically media would be mixed and cells would be inoculated.

We have rephrased this as suggested by the reviewer to "the liquid Cary-Blair medium was mixed in tryptic soy broth (TSB)" 

L163 – The addition of numbers here is much better and frames the study nicely

We thank the reviewer for this comment.

L170 – reword “could be identified”

Revised according to the suggestion of the reviewer

L184 – reword “Table 2). A small”

Revised according to the suggestion of the reviewer

L192/ 194 – 14 written as a number

Revised according to the suggestion of the reviewer

L197 – reword “hypervariable region and the flanking sequence”. Approximately how much in bp?

Approximately 2125 bp of the flanking sequence was extracted. We have rephrased this sentence with this added information.

L197/198 – “genes PilV and rci” – if these are genes, then they need to be written in italics – pilV?

Revised according to the suggestion of the reviewer

L222 – reword “contained plasmids from Japanese”

Revised according to the suggestion of the reviewer

L225 – reword “did not provide”

Revised according to the suggestion of the reviewer

L231 – reword “[15]. All of the studies used”

Revised according to the suggestion of the reviewer

L245 – reword “collections from three different studies we screened relatively large amounts”

Revised according to the suggestion of the reviewer

Table S1 – The title is still too long and cumbersome. I would suggest a shorter informative title and then make use of a legend below the table to supply extra details.

We agree with the suggestion of the reviewer and changed the title to:

"Supplementary table S1. Summary of screened cases and sequenced and selected isolates"
